

# PHIPS-HALO: the airborne Particle Habit Imaging and Polar Scattering probe – Part 2: Characterization and first results

Martin Schnaiter[1], Emma Järvinen[1], Ahmed Abdelmonem[1], and Thomas Leisner[1]

[1]Karlsruhe Institute of Technology, Hermann-von-Helmholtz-Platz 1, 76344 Eggenstein-Leopoldshafen, Germany

*Correspondence to:* M.Schnaiter (martin.schnaiter@kit.edu)

**Abstract.** The novel aircraft optical cloud probe PHIPS-HALO has been developed to establish clarity regarding the fundamental link between the microphysical properties of single atmospheric ice particles and their appropriated angular light scattering function. After final improvements have been implemented to the polar nephelometer part and the acquisition software of PHIPS-HALO, the instrument was comprehensively characterized in the laboratory and was deployed in two aircraft missions targeting cirrus and Arctic mixed-phase clouds. This work demonstrates the proper function of the instrument under aircraft conditions and highlights the uniqueness, quality, and limitations of the data that can be expected from PHIPS-HALO in cloud-related aircraft missions.

## 1 Introduction

The interaction of shortwave solar radiation with ice particles is an important process in the atmosphere, which redistributes solar light before reaching the ground. Therefore, the knowledge of the angular light scattering behavior of atmospheric ice particles is crucial for a reliable calculation of the shortwave radiative transfer in climate models and for retrieving cloud bulk properties form satellites. This is especially true for cirrus clouds that are solely composed of ice particles. Airborne in-situ investigations of cirrus clouds over the last two decades have revealed a wealth of different ice particle sizes, shapes, and crystal complexity. These microphysical data were collected with optical imaging probes that capture individual ice particles with different degrees of resolution (see Baumgardner et al. (2017) for a recent compilation and discussion of these probes). While there is now a good overview of the microphysical properties of atmospheric ice particles – at least for sizes larger than $100\,\mu\text{m}$ – their radiative properties are by far less well known because the measurement and the modeling of light scattering by complex atmospheric ice crystals are challenging.

Laboratory and modeling studies have shown that the angular light scattering properties of randomly oriented complex ice particles strongly differ from those of pristine crystals (Ulanowski et al., 2006; Smith et al., 2015; Schnaiter et al., 2016; Baum et al., 2010; Baran, 2012). While pristine hexagonal ice crystals show the $22°$ and $46°$ halo features as well as the ice bow feature at scattering angles between about $135°$ and $160°$, the angular light scattering of complex crystals is characterized by a flat and featureless function with larger scattering cross sections for side and backscattering directions. Especially the higher backscattering behavior has significant consequences for the radiative impact of cirrus clouds as more solar radiation is back-reflected to space in case of complex ice particles.



In-situ measurements of the angular light scattering function of ice particles are sparse. Angular light scattering function measurements in mid-latitude cirrus clouds as well as Arctic mid-level ice and mixed-phase clouds by the Polar Nephelometer PN (Gayet et al., 1997) revealed high back reflection (expressed by a low asymmetry parameter $g$), indicating ice crystals with significant structural complexity (Gayet et al., 2004; Jourdan et al., 2010). However, these light scattering measurements, which

are averages over thousands of ice particles, cannot easily be related to the ice microphysics even if supporting measurements are available (Shcherbakov et al., 2006; Jourdan et al., 2010). Therefore, the question what fundamentally defines the angular light scattering of individual ice particles cannot be answered by such measurements and a different approach is required. To experimentally address this question, *single* atmospheric ice particles have to be *simultaneously* measured for both, the microphysics and the corresponding angular light scattering properties.

The Particle Habit Imaging and Polar Scattering probe PHIPS-HALO has been developed to fulfill this requirement. The basic design and operation idea of PHIPS-HALO was presented in the first part of this two paper series (Abdelmonem et al., 2016), denoted by "Part 1" hereafter. In the present paper, the characterization of the instrument in the laboratory is presented (Sec. 2) followed by first results from aircraft deployments in cirrus and Arctic mixed-phase clouds (Sec. 3). The findings are summarized and an outlook is given in Sec. 4.

## 15  2   Characterization

The basic concept of PHIPS-HALO is comprehensively described in Abdelmonem et al. (2011) and in Part 1 of this paper. In brief, PHIPS-HALO is composed of two main parts, a polar nephelometer and a stereo imager (Fig. 1). A trigger detector combines these two otherwise independent systems, which enables the simultaneous acquisition of stereo images and scattered light intensities from the same particle. A comprehensive characterization of the trigger detector, the polar nephelometer,

and the stereo imager are presented in the following section together with a characterization of the temporal behavior of the detection electronics as well as the data processing and storing software.

### 2.1   Light scattering detection system

#### 2.1.1   Trigger detector

The trigger detector comprises a two-lens optical system consisting of convex lenses with 25 mm and 40 mm focal lengths

and a 7 x 0.25 mm acrylic glass (PMMA) fiber bundle that is connected to the first channel of the multi-anode photomultiplier array (MAPMT). The trigger optical system is located at a polar angle of 90° with respect to the scattering plane of the polar nephelometer, but opposite to its detector ring (Fig. 1). The trigger is azimuthally tilted by 32° out of the scattering plane with a radial distance of the front lens to the laser beam of 24.5 mm. The two-lens system produces an about two-third reduced image of the fiber bundle end on the laser beam resulting in a field of view (FOV) of the trigger detector of 0.52 mm. Projecting

this FOV into the scattering plane, which is perpendicular to the particle trajectory, results in an elliptical FOV with long and short diameters of 0.98 mm and 0.52 mm, respectively. With this the sensing area $A_{sa}$ of the instrument can be calculated to



be $A_{sa} = 0.004\,\text{cm}^2$. Here, it is important to note that the laser beam shape is Gaussian with a measured $1/e^2$ diameter of $d_l = 0.8\,\text{mm}$ and a circularity of $0.93$ at the position of the sensing area. This means that the light intensity is non-uniform across the sensing area, which consequently results in a position dependent trigger sensitivity. This is disadvantageous for sizing and counting particles based on the trigger detector signal. To improve this situation a beam shaping optical element that produces a top hat intensity profile at the sensing area has been designed and will be integrated for future measurements (see Sec. 4 for details). With $A_{sa}$ and assuming a typical airspeed of $200\,\text{ms}^{-1}$ during measurements onboard HALO, a volume sampling rate of $80\,\text{cm}^3\text{s}^{-1}$ is calculated.

The sensing area of PHIPS-HALO was also measured using uniformly sized single droplets form a piezo electric droplet generator (GESIM GmbH, Grosserkmannsdorf, Germany) mounted on a x-y-z stage. Droplets with a diameter of roughly $80\,\mu\text{m}$ where dispensed at a rate of $1\,\text{Hz}$ while slowly moving the dispenser through the sensing area along and across the laser beam. At the same time the signal of the trigger detector was monitored by an oscilloscope. The horizontal and vertical extensions of the sensing area, $\Delta x$ and $\Delta y$, were defined by the positions where the trigger signal was reduced to a tenth of its value in the center of the area. With this procedure an extension of $\Delta x = 0.53\,\text{mm}$, $\Delta y = 0.45\,\text{mm}$ was measured resulting in a reduced sensitive area of $A_{sa} = 0.0018\,\text{cm}^2$ and, consequently, in a reduced volume sampling rate of $36\,\text{cm}^3\text{s}^{-1}$ compared to the theoretical value. While the $\Delta x$ measurement is in good agreement with the theoretical value, the $\Delta y$ measurement is only about half of the theoretical value, which is a consequence of the Gaussian laser beam profile for this direction.

With these values the sensing area coincidence probability, $P_{sa}$, can be calculated by means of Poisson statistics. The average number of particles residing in the sensitive volume of the instrument is $\lambda = n \cdot A_{sa} \cdot d_l$, with $n$ the cloud particle number concentration. The probability for having more than one particle in the sensing volume at the same time is then $P_{sa}(x > 1, \lambda) = 1 - (1 + \lambda) \cdot exp(-\lambda)$. This probability is less than 1% up to particle number concentrations of $n = 480\,\text{cm}^{-3}$ and $n = 1000\,\text{cm}^{-3}$ for the theoretical and measured mapped sensing area, respectively.

When a particle is entering the laser beam at the position of the sensing area, part of the scattered light is collected by the trigger detector optics and is guided to first channel of the MAPMT, where the signal is processed and analyzed. If the trigger event is classified to be a real particle event, the data acquisition is eventually initiated. Details on the trigger signal detection and processing can be found in Part 1.

### 2.1.2 Polar nephelometer

Light scattered by individual cloud particles located in the sensing area of PHIPS-HALO is collected by a ring of 20 off-axis parabola mirrors placed at equidistant angular distances of $8°$ in the angular range from $18°$ to $170°$ at a radial distance of $83\,\text{mm}$ (see Part 1 for details). The off-axis mirrors couple the light into twenty $0.6\,\text{mm}$ PMMA fibers which guide the light to the MAPMT detector of the instrument. As reasoned in Part 1, the original concept of an additional $1°$ to $10°$ measurement at $1°$ resolution is not feasible, and, therefore, these channels are no longer used. The MAPMT consists of an array of 32 photomultipliers with anode dimensions of $0.8\,\text{mm}$ width and $7\,\text{mm}$ length. The individual channels are separated by a distance of $1\,\text{mm}$. In Part 1 a residual optical crosstalk between the individual channels of the polar nephelometer in the range of 15% to 20% was revealed by optical engineering calculations and laboratory characterizations (see Sec. 2.1.1 of Part 1). This crosstalk




could be clearly attributed to the fact that the numerical aperture (NA) and the diameter of the PMMA fibers were too large in combination with the minimum distance to the anode array of the MAPMT constrained by the $1.5\,\mathrm{mm}$ thickness of the MAPMT protection window. To solve this crosstalk problem the following redesign of the fiber-to-MAPMT coupler was performed.

(a) The $0.6\,\mathrm{mm}$ single PMMA fiber channels were replaced by fiber bundles consisting of $19 \times 0.25\,\mathrm{mm}$ PMMA fibers. The individual fibers of the bundle were arranged in a row-like manner at the coupler side in order to (i) uniformly distribute the light across the adjacent anode slit and (ii) get a fiber row width that is significantly smaller than the anode width.

(b) An array of 32 specifically designed gradient index cylinder lenses were placed between the ends of the fiber rows and the MAPMT protection window. These gradient index lenses (GRINTECH, Germany, model GT-LFCL-100-024-50-NC) have a rod-like shape with a length of $14\,\mathrm{mm}$, a width of $1\,\mathrm{mm}$ and a height of $2.37\,\mathrm{mm}$. The lenses possess anti-reflective coatings on the entrance and exit surfaces, optimized for 532 nm wavelength, and titanium coatings on the long side surfaces. The latter optically isolate the lenses when densely stacked in the coupler. The variation of the refractive index along the width of the lenses ensures in combination with their height an optimal focusing of light exiting the fiber ends into MAPMT anodes provided the fiber rows and the index lenses are precisely positioned.

Before the coupler was manufactured, comprehensive optical engineering simulations had been performed to define the optimal distances between the fiber ends and the index lenses as well as between the index lenses and the MAPMT protection window. These calculations were performed by using the optical engineering software FRED (Photon Engineering, LLC, USA). Fig. 2a shows the optical model used in these calculations. The best modeling result was achieved by assuming distances of $0.6\,\mathrm{mm}$ and $0.15\,\mathrm{mm}$ between the end face of the fibers and the entrance surface of the lenses and between the exit surface and the top surface of the protection window, respectively. The resulting irradiation plot in the plane of the anode slits is shown in Fig. 2b. With this arrangement the gradient index lenses confine the exiting light from the fiber channels to a narrow area with a width of $< 0.5\,\mathrm{mm}$ that is clearly smaller than the width of $0.8\,\mathrm{mm}$ of the anode slits. Further, the use of fiber bundles with a row-like arrangement at the MAPMT side uniformly distributes the light over a substantial part of the $7\,\mathrm{mm}$ length of the anode slits. This can be clearly seen in Fig. 3 where the irradiation profiles along and across the anode slits of the MAPMT are shown. According to Fig. 3, about 50% of the anode slit is uniformly illuminated with a gradual decrease of the irradiation towards both ends of the anode. Across the width of the anodes the cylindrical lenses nicely focus the light into the anode area without overlap between the individual focal spots.

The coupler was then characterized in the laboratory. To take into account the NA=0.33 of the off-axis parabola mirrors, which is significantly smaller than NA=0.39 of the the PMMA fibers, the following procedure was applied in this characterization. Only one fiber at a time was connected to the $18°$ parabola mirror while the connectors of the others were blocked by light tight caps. Water droplets were generated by a pump spray bottle in the vicinity of the instrument inlet and were accelerated to a speed of about $20\,\mathrm{m/s}$ by applying vacuum to the instrument outlet. Scattering data of the droplets was acquired for a short period before the setup was switched to the next fiber. This procedure was repeated until all fibers had been connected. The result of this characterization is shown in Fig. 4 where the normalized distribution of the measured intensity in the connected scattering channel ($n$) and in the neighboring channels ($n \pm 1$) is shown and contrasted to the crosstalk characterization of the





old fiber coupler. The result shows good agreement with the predictions from the optical engineering calculations (Fig. 3) and shows a clear improvement compared to the old fiber coupler. The remaining mean crosstalk to the neighboring $n \pm 1$ channels is within the range of the electrical crosstalk of 3% that is specified by the manufacturer.

The individual fiber bundles are assembled in a densely packed manner in stainless steel ferrules at the side of the parabola mirrors resulting in a clear aperture of $1.25\,\text{mm}$ diameter. This diameter together with the fiber to mirror and mirror to scattering center distances results in a FOV of $5.5\,\text{mm}$ for the individual channels. The sensing volume of the nephelometer channels is then the intersection of this FOV with the laser beam, resulting in a (skewed) cylindrical volume. While the length of the sensing volume is equal to the FOV diameter for the $90°$ channel, the length is increasing with increasing angle to the forward and backward directions. This results in a significant larger detection volume for the small and large angle scattering channels compared to the volumes of the side scattering channels (Tab. 1). While this is not a problem for cloud particle concentrations that are expected for cirrus clouds (i. e. up to $10\,\text{cm}^{-3}$), a considerable coincidence probability up to $50\,\%$ can be expected for the small and large angle scattering channels in case of particle concentrations typical for warm and mixed-phase cloud conditions (i. e. 100 to $1000\,\text{cm}^{-3}$).

Like in Part 1, the response of the improved polar nephelometer to individual particles passing the sensing volume was first characterized with NIST-traceable dry polystyrene divinylbenzene particle standards (DRI-CAL$^{\text{TM}}$ DC-50, Duke Scientific) with a nominal diameter of $49.7 \pm 2.0\,\mu\text{m}$, a standard deviation of $3.4\,\mu\text{m}$, and a refractive index of 1.59 ($\lambda = 589\,\text{nm}$). The polystyrene spheres were aerosolized in the vicinity of the instrument inlet and were transported through the optics head by applying vacuum to the outlet. Figure 5 shows the measured angular scattering functions of 20 individual polystyrene particles. The corresponding mean angular scattering function is compared to the mean theoretical function that was calculated using Mie theory and the cross-sectional equivalent diameter deduced from the images. For this, the in-house programmed MATLAB$^{\text{TM}}$ code was used, which was adopted from the Mie code printed in Bohren and Huffman (2007), to calculate the angular scattering function of the individually imaged polystyrene particles. The calculated scattering functions were then integrated over the solid angles of the polar nephelometer channels. The averaged measured scattering function agrees reasonably well with the averaged function from the Mie simulations given the uncertainties in the particle size determination, the refractive index of the particle material, and the position of the particle in the laser beam, i. e. the incident light intensity. A significant offset of the measured intensity is observed in the $100°$ to $150°$ angular range which might be a consequence of a slightly structured (roughened) surface of the polystyrene spheres in contrast to the Mie model which assumes a smooth surface. Here it is important to note that no corrections have been applied to the measured scattering signals which indicates rather similar transfer functions for the individual nephelometer channels.

In a next step individual droplets from the droplet dispenser (cf. Sec. 2.1.1) were analyzed in the same way as described for the polystyrene standards. The result of this characterization is shown in Fig. 6. An even better agreement between measurement and simulation is observed, which is a consequence of a better constrained refractive index for liquid water droplets and a more stable position of the particle trajectory through the laser beam. Specifically the steep increase of the measured scattering intensity towards the forward direction, its minimum between $80°$ and $120°$, and the about 10 times edge-like increase at the rainbow angle is nicely mimicked by the Mie simulation. From Fig. 6 clear systematic biases can be detected for the channels



82°, 90°, and 122° that will be corrected hereinafter. The bias at 122° results in the suppression of the secondary rainbow feature.

## 2.2 Imaging system

As detailed in Part 1, PHIPS-HALO consists of two camera telescope assemblies (CTA) that are arranged in a way to acquire

stereo-microscopic images of the detected cloud particle. The magnification of the zoom lenses can be manually set in the range from $1.4\times$ to $9\times$, which corresponds to field of view dimensions ranging from $6.27 \times 4.72\,\mathrm{mm}^2$ to $0.98 \times 0.73\,\mathrm{mm}^2$, respectively. This allows image acquisition either at the same or at two different field of views, which is advantageous to avoid a complete failure of the imaging system in case of alignment drifts during flight and for cloud situations when small and large ice particles are being present at the same time, e.g. in the case of small frozen droplets and large ice crystal aggregates in deep

convective outflows (Stith et al., 2014). Consequently, the optical resolution limit is dependent on the magnification setting and ranges between $7.2\,\mathrm{\mu m}$ and $2.35\,\mathrm{\mu m}$ for the $1.4\times$ and $9\times$ magnification, respectively. Using magnifications of $4\times$ and below result in an oversizing of smaller cloud particles with sizes below about $30\,\mathrm{\mu m}$ that had already been identified in Part 1 for glass beads. A correction method for this oversizing is described in Sec. 2.2.1 together with the strategy used to align the CTA focal distance for aircraft speeds in the laboratory. As there is a large difference between the maximum acquisition rates of the

CTAs ($20\,\mathrm{Hz}$) and the polar nephelometer ($6\,\mathrm{kHz}$) precautions were taken to have a robust image acquisition procedure in order to get an unambiguous assignment of the stereo-microscopic images to the corresponding scattering functions. These efforts are presented in Sec. 2.2.2.

### 2.2.1 CTA characteristics and adjustment

Figure 7 shows a comparison of cloud particle sizes deduced from the images of the stereo-microscopic imager. The image

processing algorithm that is used to deduce microphysical particle properties is described in Schön et al. (2011). The data analyzed for Fig. 7 originates from a cloud chamber experiment where a mixed-phase cloud was generated at a temperature around $-20°\mathrm{C}$. Details on the experimental procedure applied in such cloud chamber runs can be found in Vochezer et al. (2016). Hence, the data comprises small supercooled liquid droplets clustered around $20\,\mathrm{\mu m}$ in Fig. 7 as well as larger ice particles. In this experiment the CTAs of PHIPS-HALO were set to magnifications of $4\times$ and $6\times$ for CTA1 and CTA2,

respectively. The scatter plot (upper panel of Fig. 7) shows an oversizing of CTA1 compared to CTA2 for particle sizes smaller than about $30\,\mathrm{\mu m}$, which is more clearly reflected in the relative size difference (CTA1-CTA2)/CTA2·100% shown in the lower panel of Fig. 7. Thus the difference in the optical resolution limits of $\sim 5.3\,\mathrm{\mu m}$ ($4\times$) and $\sim 3.5\,\mathrm{\mu m}$ ($6\times$) results in an enhanced blur in the $4\times$ images and, consequently, in an oversizing of the particles by the image processing algorithm. Fitting an empirical function to the relative size difference allows for the size correction of the particles imaged by CTA1.

As already discussed in Part 1, the CCD cameras used in PHIPS-HALO have a trigger latency of $3.9\,\mathrm{\mu s}$ but this latency is very precise with a jitter of only $\pm 30\,\mathrm{ns}$. Hence, the particle will move a significant distance through the FOV and the depth of field (DOF) of the CTAs before the images are taken. The illumination flash is generated $4\,\mathrm{\mu s}$ after the shutter open trigger has been sent to the cameras. This means that at typical aircraft speeds of $80\,\mathrm{m/s}$ to $250\,\mathrm{m/s}$, the particle moves a distance of up to



$\sim 1\,\text{mm}$ along the instrument axis of PHIPS-HALO before the images are taken. Now, the CTAs view the particle trajectories along the instrument axis under an angle of $60°$, which means that the particle movement projected into the focal plane of the CTAs is contracted by $\sin(60°) = 0.87$. Therefore, even at fastest aircraft speeds and highest magnification the moved particle is still within the FOV of the CTA provided it is triggered on its entrance side. Knowing these technical circumstances,

the following CTA alignment procedure was developed that is schemed in Fig. 8. Starting point of this procedure is an ideal overlap between the laser beam and the FOVs of the trigger optics, the polar nephelometer optics, and the CTAs (Fig. 8a). Then the FOV of both CTAs are slightly moved along the instrument axis so that the trigger FOV is located on the entrance side of both CTA FOVs, which corresponds to a displacement of about $0.5\,\text{mm}$ (Fig. 8b). A specifically designed alignment aid is used for this task that slides into the instrument inlet tube. The alignment aid can be equipped with different specimens

(e.g. a tungsten needle) that can be moved in x-,y-, and z-direction by micrometer screws with a resolution of $10\,\mu\text{m}$. In order to mark the position of entrance side of the trigger sensitive area, the trigger optics were back-lit while a tungsten needle was slowly moved along the instrument axis until the tip is shinning up at the front edge of the trigger FOV. In this situation, the tip of the tungsten needle should be located on the inner (entrance) side of the FOV of both CTAs (Fig. 9a). With the delay for the illumination flash, which has to be set to at least $4\,\mu\text{s}$ to ensure the camera shutters are open before the flash laser is

triggered, and the expected typical airspeed, e. g. $250\,\text{m/s}$, the tungsten needle can be accordingly moved along the instrument axis to mark the particle position at the time when the flash laser is triggered (Fig. 9b). The focal plane of the CTAs can then be adjusted to this depth (Fig. 9c). This alignment procedure can be cross-checked in the laboratory using slowly moving water droplets (cf. Sec. 2.1.2) and setting the camera shutter and illuminating trigger delays to values large enough so that the particle moves the same displacement as expected under fast flight conditions. Fig. 9c gives an example stereo image that was acquired

during such a laboratory test, showing that the presented alignment procedure put the focal planes of the CTAs to the correct depth.

Depending on the magnification settings, the CTAs cover different FOVs and DOFs, resulting in different volumes that are captured in the images. For typical magnification settings of $4\times$ and $6\times$ observation volumes of $0.9\,\text{mm}^3$ and $0.3\,\text{mm}^3$ result, respectively, which is small enough to avoid any significant particle coincidence up to particle number concentrations of about

$200\,\text{cm}^{-3}$.

### 2.2.2   Assignment of the images to the corresponding angular scattering functions

As comprehensively described in the Part 1 paper, the image acquisition is initiated by the detection of a valid particle event in the signal generated by the trigger optics. Trigger pulses are then generated to start the scattering signal acquisition and - with a pre-defined delay - initiate the image acquisition, i. e. the camera shutter opening. A delay in the camera shutter trigger might

be required in situations when a significant reduction of the particle speed is expected, e. g. when checking the CTA alignment on the ground with slowly moving particles (cf. Sec. 2.2.1). As the camera shutters have a fixed latency time of $3.9\,\mu\text{s} \pm 0.03\,\mu\text{s}$ a second trigger pulse is generated by the FPGA and sent to the illumination laser at a fixed delay of $4\,\mu\text{s}$ with respect to the camera shutter trigger. A sent camera trigger flag is registered in the corresponding scattering data set, i. e. corresponding to the initial scattering data acquisition trigger. However, to avoid camera trigger registrations in situations the cameras are still





busy with the acquisition of the last image, the camera busy signals are routed to the FPGA. If either or both busy signals are high no camera trigger is sent and registered. Further, to give the instrument computer enough time to receive and store the stereo images, an additional post-trigger period can be defined by the user within that camera triggers and registrations are suppressed. With the current hard- and software, a post-trigger period of $300\,\mathrm{ms}$ has to be set in order to avoid image losses.

With these precautions there are usually as many camera trigger registrations in the scattering data set as number of stereo images stored on the solid state disk, which makes the image assignment straight forward. Occasionally, there is a mismatch between these numbers, which makes the assignment of the stereo images to the corresponding scattering function more challenging. However, also in these situations an assignment of at least a part of the acquired images becomes possible by a procedure, where the time stamp differences between all scattering data sets with a camera trigger register ($\Delta t_s$) are compared

to the time stamp differences between all acquired stereo images ($\Delta t_i$). If the difference of these two differences ($\Delta tt = \Delta t_s - \Delta t_i$) match within a reasonable limit ($\pm 50\,\mathrm{ms}$), a stereo image is assigned to a scattering data set. Otherwise either a stereo image is missing ($\Delta tt < -50\,\mathrm{ms}$) and a padding element is inserted in the camera data array, or, in case a scattering data set is missing ($\Delta tt > +50\,\mathrm{ms}$), the stereo image is excluded.

## 2.3  Electronics

PHIPS-HALO possess a set of specifically designed electronics boards for signal detection, signal conditioning and conversion, and acquisition control. Brief introductions of these boards in conjunction with a detailed step-by-step description of the signal detection and processing sequence can be found in Part 1. For the characterization presented in this section, it is important to resume the backplane controller board, which consists of a 48 MHz system controller FPGA (Xilinx, model Spartan 2) and a 16k by 18 bit FIFO memory (IDT72V265) to buffer the scattering data prior to being read by the USB daughter board

(Bitwise Systems, model QuickUSB) for transferring the data to the instrument computer (ADL Embedded Solutions, model ADL945PC-T7400). Although the FIFO is 18 bit wide, only 16 bit are used to define one word. Prior to the start of the data acquisition, the FPGA is programmed by sending a .bit file from the instrument computer to the FPGA. This program then handles the data acquisition and buffering in the FIFO as detailed in Part 1. Each particle data set consists of 38 words, so the FIFO can buffer a total of 421 particle data sets. While the polar nephelometer part of PHIPS-HALO has a data acquisition

rate of the order of several kHz, the image acquisition system, i. e. the CCD cameras, have a maximum (specified) acquisition rate of only $20\,\mathrm{Hz}$. Therefore, in order to not loosing scattering data sets, that might have corresponding images, the FPGA stops particle data acquisition in situations when the particle rate is high and the FIFO might be completely filled up before it is emptied by an USB read. Communication between the backplane board and the instrument computer is established by the USB daughter board that comes with a software library for developing custom applications. A new data acquisition software,

composed of a scattering data read application and an image receiving and storing application was developed based on the QuickUSB library and the library that comes with the CCD camera.

To characterize the temporal behavior of the detection electronics as well as the data processing and storing software, a pulse generator (Pico Technology, PicoScope model 6000) was used to generate microsecond pulses or pulse patterns at an adjustable rate. This signal was fed into the trigger channel of the signal processing and analog-to-digital boards of PHIPS-





HALO. From the timer settings for the asynchronous read process of the FIFO, a theoretical maximum acquisition rate (i. e. maximum particle rate that can be continuously detected without gaps) of 13 kHz can be calculated. However, due to a limited resolution of the software timer and the additional time need for pre-processing the FIFO output (i. e. cropping padding words) as well as the data transfer, the actual maximum acquisition rate might be lower. Therefore, the maximum rate was determined

by a step-wise increase of the set pulse rate until gaps were identified in the measured data set. This was first done without the image acquisition software running to avoid any additional delays by simultaneously handling the image transfer. A maximum continuous scattering data acquisition rate of $12\,\mathrm{kHz}$ was determined only intermittent for about $140\,\mu\mathrm{s}$ when the FIFO was read and transferred via the USB daughter board (set to $32\,\mathrm{ms}$ in these tests). In a second test also the image acquisition software was running during the pulse rate measurements over a fixed period of $20\,\mathrm{s}$. The result of this test is depicted in Fig. 10 where

the time difference between consecutive data sets is plotted for three different pulse rates. While for 2 Hz and 2 kHz cases the time difference is constant and equal to the inverted frequency, the time analysis of the scattering data set for 6 kHz shows infrequent delays between consecutive data sets up to $100\,\mathrm{ms}$. A total of 55 data sets have a time difference longer than the $0.167\,\mathrm{ms}$ defined by the set pulse rate. Consulting the number of acquired images of 67, it is obvious that the image acquisition process holds the scattering data read process until the acquired images (two per camera trigger) have been transferred to the

solid state disk. This often results in a full FIFO, a stop of scattering data acquisition, and, consequently, in a loss of data sets. Further investigations revealed a maximum particle acquisition rate of $3.5\,\mathrm{kHz}$ without any loss in the scattering as well as image data. Using the volume sampling rate of $36\,\mathrm{cm}^3\mathrm{s}^{-1}$ at $200\,\mathrm{ms}^{-1}$ airspeed (Sec. 2.1.1), this facilitates a maximum particle concentration of about $100\,\mathrm{cm}^{-3}$ that can be continuously acquired without any loss.

In a final test, the acquisition dead time after individual trigger events were investigated by applying a train of double pulses

with decreasing time lag. This test revealed a dead time of about $12\,\mu\mathrm{s}$ that is required by the electronics for pulse analysis and analog-to-digital conversion of each of the 32 channels followed by a successive read of the converted signals by the FPGA. Assuming again an airspeed of $200\,\mathrm{ms}^{-1}$, this results in a minimum particle distance of $2.4\,\mathrm{mm}$ that can be resolved by the electronics, meaning that the post-trigger data processing is fast enough to perform a shattering analysis based on the inter-particle arrival times - at least for cirrus cases.

## 3   First results

In this section first results from aircraft measurements with PHIPS-HALO are presented. Although, the instrument was deputed on the HALO aircraft in the missions ML-CIRRUS and ACRIDICON-CHUVA in 2014, the examples that are given below are from two recent aircraft projects, namely within the ARISTO program conducted with the NSF/NCAR GV HIAPER (Gulfstream-V High-performance Instrumented Airborne Platform for Environmental Research) in February and March 2017

and the ACLOUD mission conducted with the AWI Polar 6 aircraft of Alfred Wegener Institute in May and June 2017. The reason for this confinement is that the full instrument capabilities including a crosstalk free scattering function measurement and a reliable image-to-scattering function correlation were only available in these recent missions. First results from the ACRIDICON-CHUVA mission are presented elsewhere (Wendisch et al., 2016).



The Airborne Research Instrumentation Testing Opportunity (ARISTO) is a NSF funded program for testing newly developed or highly modified instruments as part of their development efforts. The 2017 phase of the program provided testing opportunities primarily for instrumentation that will be participating in the SOCRATES (Southern Ocean Clouds, Radiation, Aerosol Transport Experimental Study) field project in 2018. The ARISTO-2017 project took place in Broomfield, CO in the period February 20 to March 10, 2017. Five research flights in clear-sky and in clouds were conducted over the Great Plains north and east of Broomfield (Colorado, Wyoming, Nebraska, Kansas). In one long test flight a marine boundary layer cloud deck west of California was probed.

The Arctic CLoud Observations Using airborne measurements during polar Day (ACLOUD) mission took place from May 22 to June 28, 2017 based in Longyearbyen (Svalbard, Norway). Within ACLOUD PHIPS-HALO was participating in 17 research flights north and west of Svalbard. In these flights PHIPS-HALO collected a unique data set of the microphysical and light scattering properties of ice in Arctic stratiform clouds.

### 3.1 Advantages of a stereo imager

Before results of the correlated microscopic and angular light scattering measurements are presented, the advantages of the stereo-imaging capabilities of PHIPS-HALO are briefly highlighted. A general problem in two dimensional optical imaging of ice crystals - even in the case of real in focus optical microscopy like used in PHIPS-HALO - there are always hidden or out of focus portions of the particle present that makes a representation of its three dimensional geometric structure impossible. In Fig. 11 two examples of skeleton plates are depicted that were sampled by PHIPS-HALO during ACLOUD in ice precipitation underneath a mid-level cloud at temperatures between $-10\,^{\circ}$C and $-14\,^{\circ}$C. The stereo-images of PHIPS-HALO reveal that these crystals are actually composed of multiple stacked skeleton plates. In the lower example shown in Fig. 11 three hexagonal plates are concentrically stacked along the basal facet, which becomes obvious by inspecting the image of CTA1 (left). Having only the image of CTA2 (right) available, the crystal would have been classified most likely as a single skeleton plate. Although, a stacked plate arrangement is identifiable in CTA2 (right) of the upper example in Fig. 11, the one side-plane that is radiating in a different direction becomes visible only by imaging the crystal under a different viewing angle as in the case of CTA1 (left).

There are further advantages of using a stereo imager, which stem from the possibility of using different magnifications for the two images. Different magnifications result in different FOVs, which enables to (a) image large and small ice particles at the same time without having the problem that large crystals are not completely captured in a narrow FOV and (b) capture coincident particles and possible break-up events of large ice particles or aggregates in a single frame.

### 3.2 Single particle angular scattering functions

The proper function of the image-to-scattering function assignment procedure presented in Sec. 2.2.2 was demonstrated by the following simple test. The PHIPS-HALO image data from a $12\,\mathrm{min}$ long flight leg in a marine boundary layer cloud conducted during the ARISTO2017 project was searched for large droplets with diameters in a narrow size range between $64\,\mu\mathrm{m}$ to $66\,\mu\mathrm{m}$. In the complete imagery only three droplets fulfill this criterion (out of 588 total images). As the instrument trigger threshold



was set to about $30\,\mu m$ and because the cloud itself was dominated by smaller droplets, the number of instrument triggers from these large droplets are very low and about 0.5% compared to the total amount of triggers in a specific time. Consequently, the scattering data of the selected images are neighbored by triggers from significantly smaller particles. Further, by selecting the two images with the largest difference in time, the robustness of the assignment method against time related influences can be

checked. The two imaged droplets with image numbers 4 and 524 have diameters of $66\,\mu m$ and $64\,\mu m$, respectively, and were captured with a time difference of almost $10\,min$. Figure 12 shows the scattering functions that were automatically assigned to these images by the assignment procedure. As expected for nearly equal-sized droplets the scattering functions show almost no difference, which is a clear proof of a successful image-to-scattering function assignment. The two scattering functions were then averaged and compared with the result of a Mie calculation for a droplet with $65\,\mu m$ diameter. Again, an excellent

agreement is achieved, which shows that the scattering function measurement is reliable also under flight conditions.

The correlated measurement of microphysical and angular light scattering properties on a single particle basis gives unprecedented research opportunities in the field of cloud physics. The correlated data can be compiled as single (ice) particle data or as habit- and phase-specific averaged data. Examples of single ice particle data will be given in the remainder of this section. Averaged data are presented Sec. 3.3.

Figure 13 gives single particle angular scattering functions measured for two plate-like ice particles during the ARISTO2017 project. These two plates were selected because (i) they have a similar size and (ii) they are similarly oriented. A detailed inspection of the stereo images reveal that the two crystals significantly differ in the degree of surface structure. As a consequence of this structural difference, the angular light scattering properties of the two crystals vastly differ with a more diffuse light scattering behavior in case of the highly structured crystal (denoted as (b) in Fig. 13) compared to the less structured crystal

(denoted as (a)). The diffuse spacial light scattering of crystal (b) results in a flat and featureless angular scattering function with significant intensity measured also for side and backscattering directions where the less structured crystal (a) shows a minimum in its light scattering behavior. Crystal (a), on the other hand, shows at least two local maxima at the scattering angles $26°$ and $154°$. These maxima are likely the result of refracted ($26°$) and internally reflected ($154°$) rays, although an analysis based on modeling results, like in the work of Shcherbakov et al. (2006), is necessary to confirm this interpretation. According to

Shcherbakov et al. (2006) a pristine hexagonal ice particle with a low degree of crystal complexity and distortion is necessary to generate such features in the oriented single particle angular light scattering function. Note that the diffuse light scattering behavior of crystal (b) is also indicated by its darker appearance in the corresponding bright field stereo images. This example again demonstrates the reliable image-to-scattering function assignment and shows the unique capability of PHIPS-HALO in terms of measuring the link between cloud ice microphysical and light scattering properties. Such single particle data of real

atmospheric ice particles should be of high value for modelers developing advanced ice particle optical models that include ice crystal distortions like hollowness, air inclusions, and surface roughness.

### 3.3 Averaged angular scattering functions

While the previously shown single particle angular light scattering functions are of high value for understanding the fundamental light scattering properties of atmospheric ice particles, cloud angular scattering functions averaged over many single particle





measurements are of more interest when it comes to the radiative impact of the clouds on the energy budget of the atmosphere. Despite the common method of a separate analysis of the microphysical and scattering ensemble data, the correlated single particle data from PHIPS-HALO facilitates unprecedented analysis methods also in case of cloud ensemble data. These novel capabilities are presented in the following two sections.

### 3.3.1 Habit-specific angular scattering functions from ice clouds

The correlated single particle image and scattering data provided by PHIPS-HALO can be used to calculate habit-specific averaged ice particle ensemble data even in situations when the cloud is not homogeneous and composed of ice crystals with different habits. To highlight this unique capability, a $\sim 30$ min flight leg from RF02, conducted on February 24, 2017 during the ARISTO-2017 project, was analyzed. In this leg, the GV aircraft probed an extensive cloud field over west and central Nebraska.

In a first analysis, bullet-rosettes were selected from the PHIPS-HALO imagery that was captured between 20:36 UT and 20:43 UT when the aircraft descended from an initial altitude of $7.7\,\mathrm{km}$ to $6.7\,\mathrm{km}$. During this descent the GV profiled a thin cirrus layer, which existed at ambient temperatures between $-47^\circ$C and $-41^\circ$C. The cloud was dominated by hollow columns and hollow bullet-rosettes with minor fractions of irregulars and rosettes with side planes. Only crystals that could be clearly classified as (hollow) bullet-rosettes, i. e. excluding crystals with side planes, were selected form the PHIPS-HALO stereo imagery, resulting in a total number of 54 selected crystals (see Fig. 14 for examples). The individual angular scattering functions assigned to this set of selected crystals are plotted on the right of Fig. 14 together with the habit-specific averaged angular scattering function of this crystal class.

In a second analysis, plate-like crystals were selected from the period 20:52 UT to 21:00 UT when a thicker altostratus cloud was profiled between $5.4\,\mathrm{km}$ and $4.4\,\mathrm{km}$ altitudes. The temperatures in this cloud were significantly warmer and ranged between $-31^\circ$C at cloud top and $-26^\circ$C at the bottom of the cloud. The ice crystal habit distribution was much broader in this cloud compared to the thin cirrus case, with plate aggregates and side planes but also with significant numbers of columns, rosettes, rimed particles, and small irregulars. Again, only crystals with a clear plate and side plane habit were selected from the PHIPS-HALO imagery, resulting in a total number of 17 selected crystals (see Fig. 15 for examples). The individual angular scattering functions assigned to these crystals and the habit-averaged ensemble scattering function are plotted on the right of Fig.15. By comparing the habit-specific averaged scattering functions of Figs. 14 and 15 it becomes obvious that these functions do not significantly differ. This indicates that particle ensembles composed of ice crystals that show a significant complexity on a single particle basis possess similar flat and featureless average angular scattering function even if their basic crystal habit differ (columnar vs. plate-like in this case).

The above type of analysis is important to answer the question which microphysical property of ice clouds dominate their angular light scattering behavior – the crystal habit or the crystal complexity in terms of distortions, inclusions, and surface roughness. Although the above examples have demonstrated that this question can be addressed by measurements with PHIPS-HALO, further detailed analyses with larger data sets are necessary to come to statistically significant conclusions. This will be the subject of future studies after PHIPS-HALO has been participated in further cloud related aircraft projects.



### 3.3.2 Phase-specific angular scattering functions from mixed-phase clouds

The correlated single particle microphysical and light scattering measurement capability of PHIPS-HALO can also be used to deduce phase-specific averaged angular scattering functions of cloud particle ensembles in mixed-phase clouds. Fig.16 gives an example of such an analysis from data collected in an Arctic mixed-phase cloud during the ACLOUD project. On June 6,

2017 the Polar-6 aircraft profiled an approximately $200\,\mathrm{m}$ thick low-level arctic stratus cloud northwest ($81°15'$ N, $9°15'$ E) of Svalbard, Norway. The temperature within the cloud ranged between $-1°$C at cloud base and $-4°$C at cloud top. The trigger threshold of PHIPS-HALO again was set in a way that the instrument started to trigger on droplets with diameters larger than $30\,\mu\mathrm{m}$. The acquired stereo imagery clearly revealed the presence of some drizzle drops with sizes up to $150\,\mu\mathrm{m}$ located near cloud top where small hollow columns were observed. The observed ice crystal habits were columnar throughout the cloud

with needles, aggregates of needles, and hollow columns – all with different degrees of surface roughness and riming.

The corresponding angular scattering functions of the imaged droplets are narrowly grouped with the primary and secondary rainbows clearly indicated at the $138°$ and $122°$ detection angles, respectively (inset of Fig.16). The droplet diameters deduced from the stereo images are used in Mie theory to calculate the averaged angular scattering function of this particle ensemble that nicely mimics the average of the measured functions. Scattering functions from ice particles are more varying but have a rather

flat angular dependence, which is likely the consequence of a significant single particle complexity in terms of hollowness, surface roughness, and riming. Consequently, the resulting averaged scattering function of the imaged ice particle ensemble shows a flat and featureless angular dependence and is clearly distinct from the corresponding function of the droplets for scattering angles larger than $50°$. Interestingly, the averaged scattering function of these mixed-phase ice particles does not significantly differ from the corresponding habit-specific functions presented in the previous section, even though those crystals

were measured in a completely different atmospheric compartment. However, further measurements and analyses are necessary to clarify whether this observation is a general property of atmospheric ice particle ensembles.

### 4 Summary and Outlook

This paper presents the final version of the novel aircraft cloud probe PHIPS-HALO. The final design improvements in the optical setup and software are presented in conjunction with a thorough instrument characterization in the laboratory. Airborne

measurements are used to show the proper function of PHIPS-HALO under flight conditions and highlight the unique data that can be gathered in cirrus and mixed-phase clouds. These first results show that new directions in the field of cloud and climate research can be followed with PHIPS-HALO. For example, the correlated single particle measurements should be of high value for modelers developing advanced ice particle optical models. First advances in this direction have already been undertaken (Stegmann et al., 2016). Moreover, the stereo imagery provides an unprecedented view into the microphysical

details of atmospheric ice particles, and the angular light scattering data have the potential to improve the ice cloud light scattering parametrization in future climate models. Therefore, PHIPS-HALO would be beneficial for future cloud-related aircraft mission.





In an ongoing work, PHIPS-HALO is being upgraded for polarization measurements in the backscattering angular region. The goal of this upgrade is to validate polarimetric satellite measurements in future aircraft projects. As part of this upgrade, beam shaping optics will be implemented in the polar nephelometer part of PHIPS-HALO in order to generate a top hat laser beam intensity across the sensitive area. With this upgrade a very defined triggering behavior can be expected for all particle

sizes that will reduce the position dependent variations in the light scattering intensity and the number of out of focus images in case of large ice crystals.

*Author contributions.* MS is leading the development of PHIPS-HALO and acquired the necessary funds. MS designed the MAPMT fiber coupler and performed the optical engineering calculations. MS and EJ did the further technical improvements, characterized PHIPS-HALO in the laboratory, deployed the instrument in aircraft projects, and analyzed the results. AA was involved in the instrument design and

assembled the first version of PHIPS-HALO. TL supported the work by additional funding. MS wrote the manuscript with comments from all co-authors.

*Acknowledgements.* The authors would like to express their gratitude to the technical crews at IMK-AAF, at the Research Aviation Facility of NCAR, and at AWI for their support in certifying and integrating PHIPS-HALO to the different airborne platforms. Dirk Kalmbach from AWI and Guillaume Mioche from LaMP, France are thanked for their technical help and their support in operating PHIPS-HALO in

ACLOUD. This work was funded within the Helmholtz Research Program Atmosphere and Climate and by the German Research Foundation (DFG grants SCHN 1140/1-1, SCHN 1140/1-2, and SCHN 1140/3-1) within the DFG priority program 1294 (HALO). The National Science Foundation (NSF) is thanked for providing access to the NSF/NCAR C-130 and HIAPER aircrafts during the ARISTO 2016 and ARISTO 2017 projects. The German Research Foundation is thanked for providing access to the AWI Polar-6 aircraft during the ACLOUD project as part of the Transregional Collaborative Research Center TR172 (AC3).





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




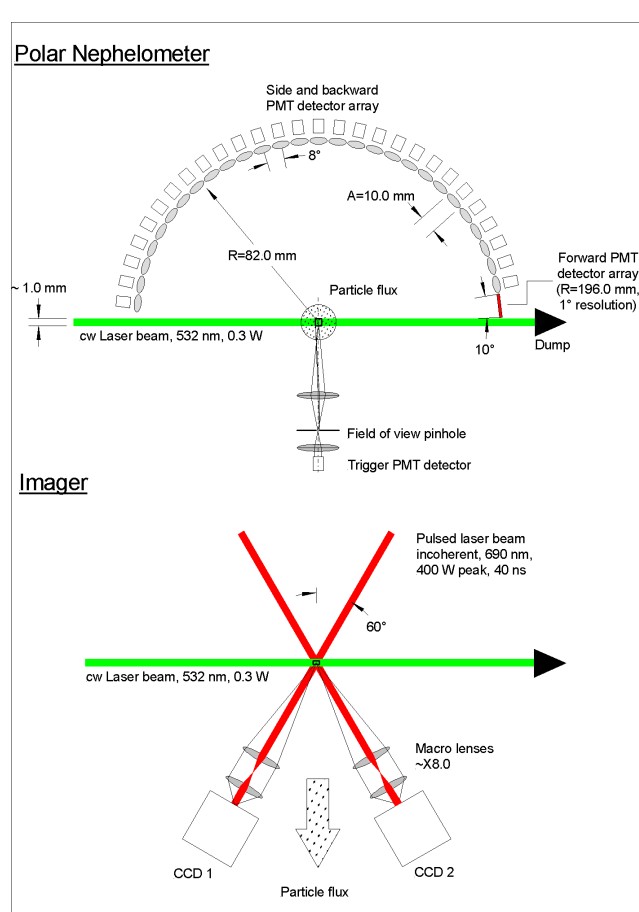

**Figure 1.** Schematic diagram of the two main components of PHIPS-HALO. The scattering plane of the polar nephelometer (upper diagram) is oriented at 90° to stereo imager (lower diagram). Note that the forward PMT detector array is no longer used. Figure adapted from Stegmann et al. (2016).





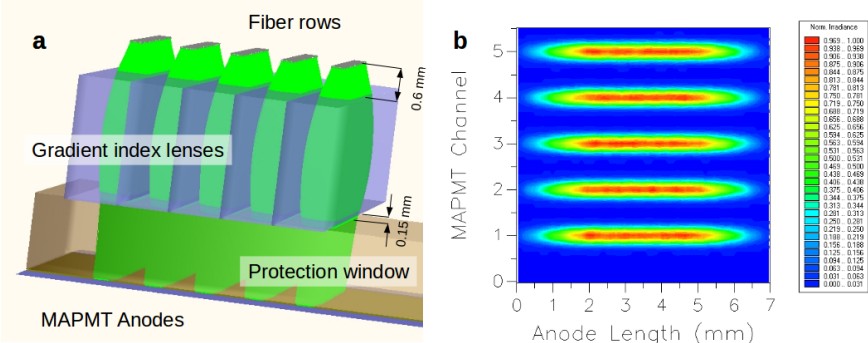

**Figure 2.** Optical engineering simulations of the redesigned fiber-to-MAPMT coupler used in the polar nephlometer part of PHIPS-HALO.
Optical model (a). Simulated irradiation in the plane of the the anode slits of the multi-anode PMT (b).





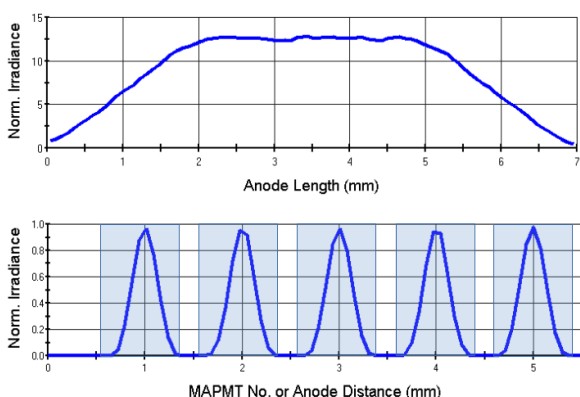

**Figure 3.** Normalized irradiation profiles along (top) and across the anode slits (bottom) of the MAPMT. The shaded areas in the lower panel indicate the width of the anodes.





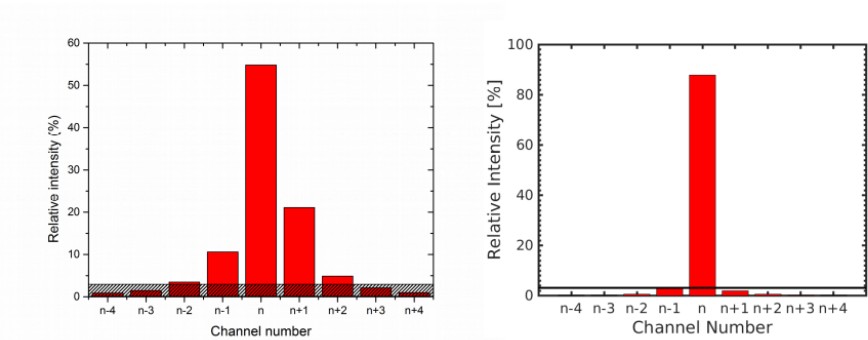

**Figure 4.** Characterization of the optical crosstalk between adjacent channels of the MAPMT with the old version (left) and the redesigned version of the fiber coupler (right). Note that the manufacturer specified electrical crosstalk level of 3% is indicated in the graphs. See text for further details.

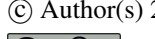



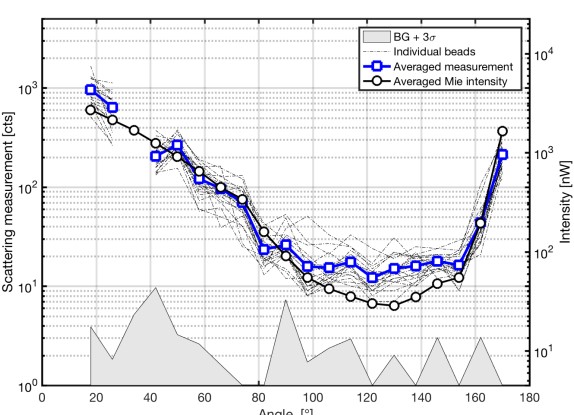

**Figure 5.** Characterization of the polar nephelometer with dry polystyrene spheres with a nominal monodisperse diameter of $49.7 \pm 2.0\,\mu\text{m}$. The image analysis results where used as input for Mie calculations. The averaged Mie scattering function is shown in black.





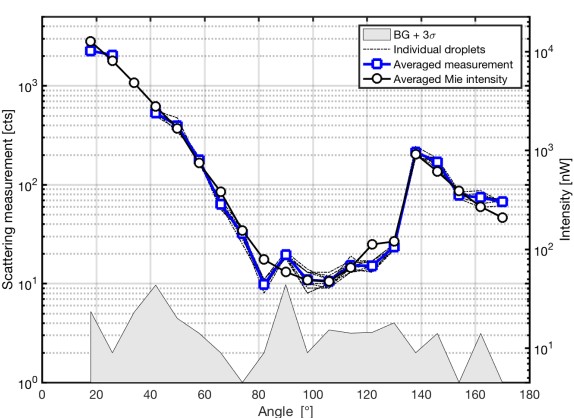

**Figure 6.** Characterization of the polar nephelometer using uniformly sized single droplets form a piezo electric droplet generator.



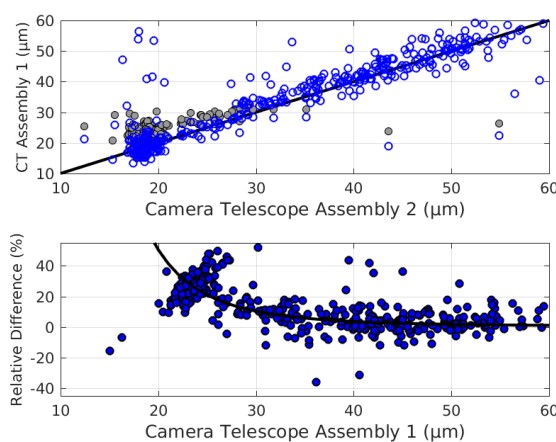

**Figure 7.** Example of an oversize correction for small particles and a low magnification setting of 4 in camera telescope (CT) assembly 1. CT assembly 2 was set to 6 fold magnification. For cloud particle sizes smaller than 30 μm, the image analysis algorithm applied to the CT assembly 1 images starts to oversize the particles due to enhanced image blur (gray symbols in the upper panel). An oversizing in the range between 10% and 50% is observed (blue symbols in the lower panel). By using the sizing results for CT assembly 2, a correction function can be deduced to empirically describe this oversizing (black line in the lower panel) and to correct the sizes deduced for CT assembly 1 (blue symbols in the upper panel).





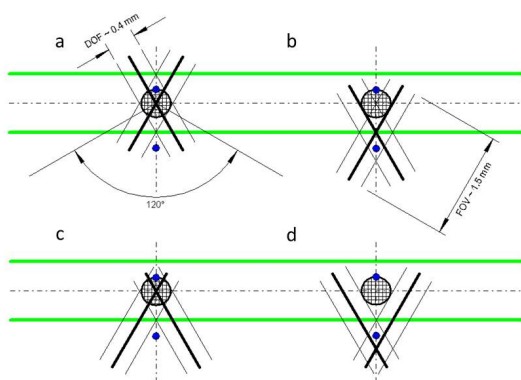

**Figure 8.** Schematics of the alignment strategy for the zoom lens focus. Ideal overlap between the the two zoom lens field of views (FOV) and the trigger sensing area (a). The trigger sensing area is represented by the hatched area centered in the laser beam diameter (the edges of the laser beam are indicated by the green lines). The positions of the particle when entering the sensing area and after the camera shutter delay are shown as blue dots, assuming an airspeed speed of 200 m/s. Note that the particle is already at the edge of the FOV and outside the depth of field (DOF) of both zoom lenses when the camera shutters open. Displacement of the FOVs relative to the sensing area by about 0.4 mm to compensate for the particle movement during the shutter delay (b). Zoom lens focus adjustment on a slowly moving particle in the laboratory (c). Focus adjustment on a fast moving particle during aircraft measurements (d).





**Figure 9.** Stereo images of the tungsten needle taken during the alignment procedure described in Sec. 2.2.1. Needle at the edge of the trigger FOV (a). Needle tip marks the particle position when the CTA shutters have been opened and the illumination flash is fired at an assumed airspeed of 250 m/s (b). Focal planes of the CTAs have been adjusted to position (depth) of the needle tip (c). Cross-check of the alignment with slowly moving water droplets in the laboratory using a illumination trigger delay scaled to the differences in air speed between laboratory and aircraft operation (d).





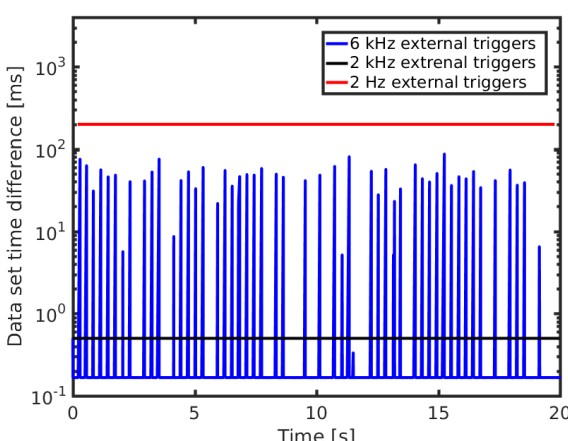

**Figure 10.** Characterization of the maximum scattering data acquisition rate of PHIPS-HALO in case of simultaneous image acquisition at 3 Hz. Note that the maximum rate without a simultaneous image acquisition is 12 kHz.



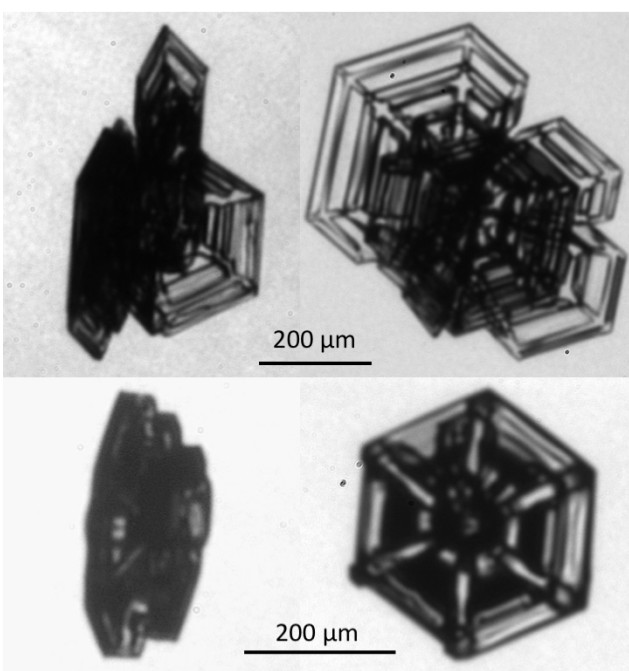

**Figure 11.** Two examples of stereo images for plate-like ice crystals captured in ice precipitation underneath an Arctic mid-level cloud. The examples nicely show the advantage of having a second view of the crystal.



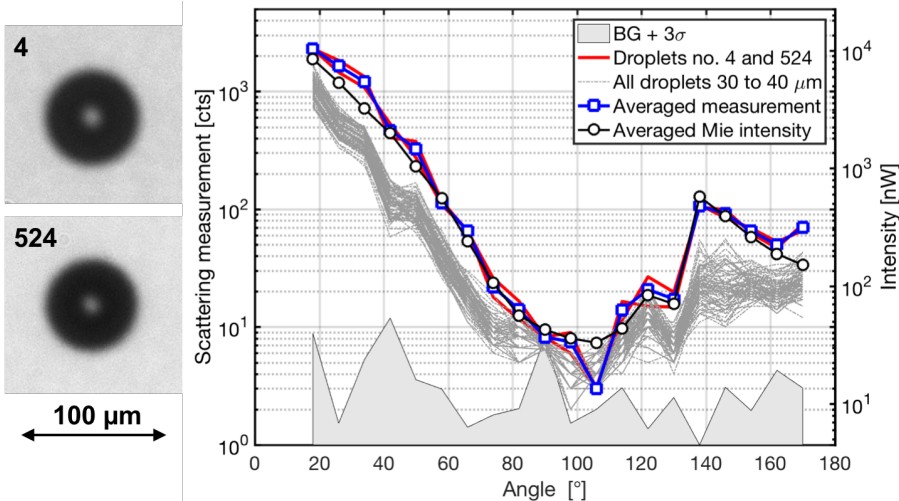

**Figure 12.** Test of the automatic image-to-scattering function assignment procedure. Sparsely imaged large diameter droplets with similar diameters of $64\,\mu m$ and $66\,\mu m$ were selected from the PHIPS-HALO imagery collected in a $12\,min$ flight leg in a marine boundary layer cloud. The images (with numbers 4 and 524 in imagery) were automatically assigned to the particle trigger events no. 738 and 3986 of the angular light scattering data set (red lines). The data set numbers correspond to a time difference about 10 minutes between the two trigger events. The averaged scattering function of these two events (blue) can be nicely reproduced by Mie calculations for a single droplet with a diameter of $65\,\mu m$ (black). Single particle angular light scattering functions assigned to those droplets from the imagery that have a diameter between 30 and $40\,\mu m$ (dash-dotted lines).





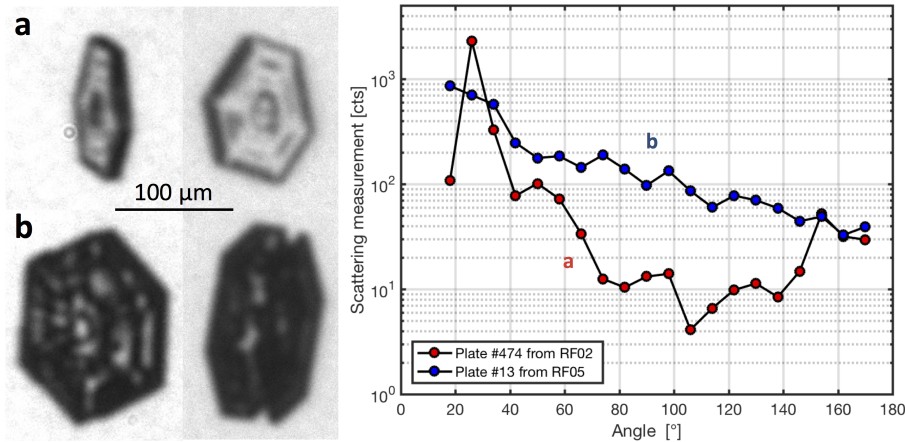

**Figure 13.** Stereo images of two plate-like crystals captured during the ARISTO2017 project (left). Maximum crystal dimensions are $132\,\mu m$ and $167\,\mu m$ for crystal (a) and (b), respectively. Correlated angular scattering functions measured for the two crystals (right). Note the $26°$ and $154°$ local maxima in case of the more pristine crystal (a).





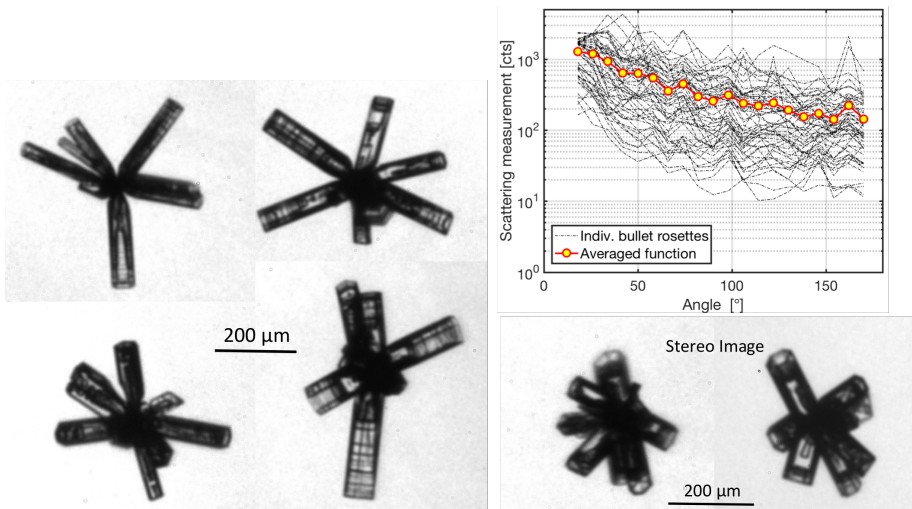

**Figure 14.** Examples of hollow bullet-rosettes selected from the imagery captured by PHIPS-HALO during the descent through a $\sim -45°$C cirrus cloud over Nebraska. Measured single particle angular light scattering functions automatically assigned to the selected hollow bullet rosettes (right, dash-dotted lines). The habit-specific averaged angular scattering function of this particle class is given in red.





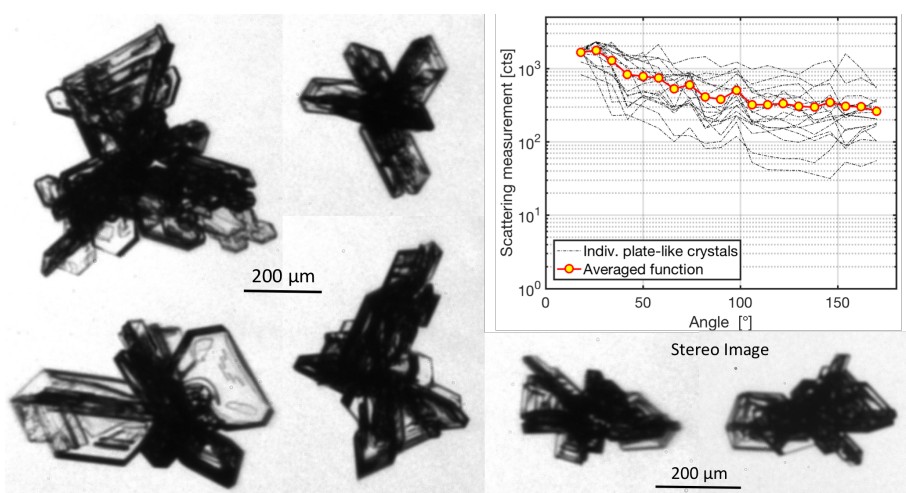

**Figure 15.** Same as in Fig. 14 but for plate-like and side plane crystals selected from the imagery captured by PHIPS-HALO during the descent through a $\sim -28°$C altostratus cloud over Nebraska.





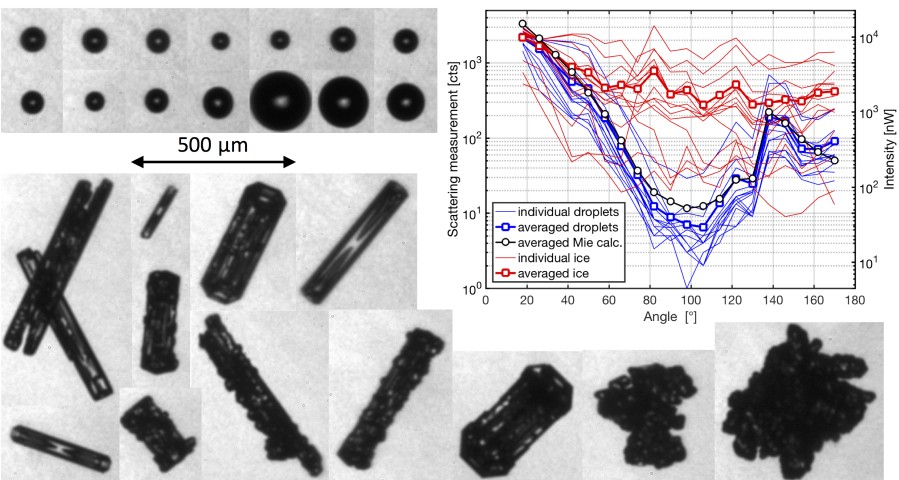

**Figure 16.** Example of PHIPS-HALO data acquired in a low-level Arctic stratus cloud. Bright field microscopic images of liquid droplets and ice particles are grouped on the left. The angular scattering functions of the corresponding droplets (blue) and ice crystals (red) are plotted on the right. Averaged scattering functions of each particle class are given in blue and and red symbols. The averaged theoretical light scattering function from Mie theory is given in black symbols. Note that the averaged function of the ice crystals (red symbols) is clearly distinct from the corresponding function of the liquid droplets (blue symbols) for scattering angles larger than $50°$.



**Table 1.** Field of view (FOV) areas, detection volumes and coincidence probabilities for the optical arrangement of the polar nephelometer of PHIPS-HALO. Note that coincidence can be neglected for cirrus but might be considered for mixed-phase cloud conditions.

| Angle | Fiber diameter | FOV Area | Detection Volume | Coincidence at $10\,\mathrm{cm}^{-3}$ | Coincidence at $100\,\mathrm{cm}^{-3}$ |
|---|---|---|---|---|---|
| ° | mm | $\mathrm{cm}^2$ | $\mathrm{cm}^3$ | % | % |
| 18 | 1.25 | 0.077 | 0.009 | 0.4 | 23 |
| 26 | 1.25 | 0.055 | 0.006 | 0.2 | 14 |
| 34 | 1.25 | 0.120 | 0.005 | 0.1 | 9 |
| 42 | 1.25 | 0.100 | 0.004 | 0.1 | 7 |
| 50 | 1.25 | 0.088 | 0.004 | 0.1 | 5 |
| 58 | 1.25 | 0.079 | 0.003 | 0.1 | 4 |
| 66 | 1.25 | 0.073 | 0.003 | 0.1 | 4 |
| 74 | 1.25 | 0.070 | 0.003 | 0.0 | 4 |
| 82 | 1.25 | 0.068 | 0.003 | 0.0 | 3 |
| 90 | 1.25 | 0.067 | 0.003 | 0.0 | 3 |
| 98 | 1.25 | 0.068 | 0.003 | 0.0 | 3 |
| 106 | 1.25 | 0.070 | 0.003 | 0.0 | 4 |
| 114 | 1.25 | 0.073 | 0.003 | 0.1 | 4 |
| 122 | 1.25 | 0.079 | 0.003 | 0.1 | 4 |
| 130 | 1.25 | 0.088 | 0.004 | 0.1 | 5 |
| 138 | 1.25 | 0.100 | 0.004 | 0.1 | 7 |
| 146 | 1.25 | 0.120 | 0.005 | 0.1 | 9 |
| 154 | 1.25 | 0.055 | 0.006 | 0.2 | 14 |
| 162 | 1.25 | 0.078 | 0.009 | 0.4 | 23 |
| 170 | 1.25 | 0.132 | 0.016 | 1.3 | 48 |