# Peer review of "PHIPS-HALO: the airborne Particle Habit Imaging and Polar Scattering probe – Part 2: Characterization and first results"

_Atmospheric Measurement Techniques, 2017_

## Referee Comment (RC1) · Anonymous Referee #1 · 20 Sep 2017

Review of "PHIPS-HALO: the airborne Particle Habit Imaging and Polar Scattering probe – Part 2: Characterization and first results" by Schnaiter et al.

Recommendation: Accept with minor revision

This paper characterizes the response of the PHIPS-HALO probe that was introduced in the Part 1 paper published last year. In addition to the laboratory characterization of the light scattering detection system, imaging system and electronics, some first results from two field campaigns in the Arctic and in the vicinity of Colorado during the ARISTO 2017 field campaign are presented. I find that the probe does offer a unique way of looking at data from having stereoscopic images and from having scattering

phase functions of individual particles coincident with the scattering properties, and will likely allow some fundamental questions describing the relation to cloud microphysics and radiation to be answered in the near future. Thus, I think its publication in AMT is appropriate. However, I am recommending a few minor revisions that I think will improve the quality and readability of the paper.

1) The paper discusses the PHIPS-HALO in isolation from any other imaging probes and scattering probes that are currently available. It would be nice to compare the advantages and disadvantages of PHIPS-HALO with some of these other probes. Perhaps a table could be constructed where some parameters of different probes that image particles and scattering phase functions could be compared (e.g., sample area, number of crystals imaged in given time period, range of particle sizes, sizes of particles detected, data volume, etc.). This would be helpful for future users of the probe.

2) I think that the authors should exhibit more caution in some conclusions that they make out of a very limited set of data. I think the current paper is very powerful at showing the types of questions the probes can answer, but less powerful at actually answering these questions given the very limited amount of data that are presented. Some of the conclusions in Sections 3.2 and 3.3.1 are especially problematic. For example, the authors claim that the images in Fig. 13 show that a highly structured crystal (b) gives flat and featureless phase functions, whereas less structured crystals (a) exhibit peaks at two specific angles. I found this less than convincing: when I compare the (a) and (b) images I only see that the image in (b) is darker, as later commented on by the author. If the b particle is indeed more structured, the authors need to show the specific places on the crystals (perhaps circled) where this structure is seen. I am also not convinced that the particles in (a) and (b) are similarly oriented. They look to me that they could be oriented with different angles. Can the authors do some simple scattering simulations to show how different orientations of the same particle affect the scattering phase function? If there is a difference in 10 degrees, for example, is this sufficient to show different scattering functions? Over what angles would

the scattering patterns be similar, and how close do the scattering functions need to be in order to be classified as similar? Similarly, I am concerned with the analysis in Section 3.3.1 where the authors make the overarching claim that "particle ensembles composed of ice crystals that show a significant complexity on a single particle basis possess similar flat and featureless average angular scattering function even if their basic crystal habit differ (columnar vs. plate-like in this case)." I think a much more thorough analysis needs to be done, including using scattering models to see how different orientations of crystals and different constructions of bullet rosettes with varying numbers of rosettes and orientations, affect the scattering properties before making such a conclusion. The authors themselves seem to explicitly acknowledge this when they stated that the "above examples have demonstrated that this question can be addressed by measurements with PHIPS-HALO, [but] further detailed analyses with larger data sets are necessary to come to statistically significant conclusions." I would recommend toning down the earlier statements, and supplying some scattering simulations, to better justify the discussions in the earlier part of the paper.

Minor Comments:

Page 3, line 1: How does the sample area (and other parameters) compare with other probes? See major comment 1.

Page 8, line 21 bits not bit

Page 8, line 26: lose not loosing

Page 9, line 19, was not were

Page 9 line26: Remove was

Page 10, line 3: I don't think ARISTO was designed to test instrumentation from SOCRATES, even though many of the experiments used in ARISTO will ultimately be used in SOCRATES

Page 10, lines 15-16: I don't think it is true that there are always portions of the particle

in other imaging probes that are out of focus. While I think it is true that some portions of the particles imaged that are out of focus, but there are individual particles that are entirely in focus. I'm not sure if this is a misinterpretation of the English that is written, but it should be noted that entire particles are in focus in other probes (though some particles are entirely out of focus).

Page 13, line 32: missions not mission

---

## Referee Comment (RC2) · M. Klingebiel (Referee) · 4 Oct 2017

**Full review of Schnaiter et al., AMT 2017 (based on the manuscript version of 30 August 2017)**

**General comments**

The manuscript „PHIPS-HALO: the airborne Particle Habit Imaging and Polar Scattering probe – Part 2: Characterization and first results" is the second part of a study presenting a novel aircraft optical cloud probe. This part is focusing on the characterization and the first measurements from the PHIPS-HALO instrument.

The unique part of the PHIPS-HALO is the combination of a polar nephelometer and a stereo imager. Both components together allow for measurements of the microphysical properties and the appropriate angular light scattering function of single particles.

In this manuscript, the authors characterize the main components of the instrument (light scattering detection system, imaging system, electronics) and present some first results from two research campaigns.
For example, the authors explain very clearly why and how they redesigned the fiber-to-MAPMT coupler inside the polar nephelometer in order to avoid optical crosstalk between adjacent channels of the MAPMT, which is an important factor for obtaining reliable measurements. For the imaging system, they introduce a correction method, which is used to correct the oversizing of smaller cloud particles in order to get adequate results.

All in all, the manuscript is very well written and has a clear structure. I would suggest the manuscript to be published after minor revision. This should address the following points:

**Major comments**

Page 9, Line 26 – 33: You mention that the instrument was used during four aircraft missions. I am wondering how the measurements from the PHIPS-HALO agree with particle measurements from other imaging instruments or another polar nephelometer. I think this is the biggest weakness of the manuscript, because the authors show results from only a single instrument. It would be a beneficial to know if the measurements of the angular scattering function are similar to measurements from another polar nephelometer. The comparison could be done by using homogenous cloud sections. If other instruments were not available on the aircraft, it might be possible to use cloud chamber studies for an instrument intercomparison.

Page 10, Line 12 – 28: You point out the advantages of the stereo imager very clearly, but the advantages of the whole PHIPS-HALO instrument in comparison to other instruments is neglected. It would be nice to have a table or a paragraph which summarizes the advantages of this novel cloud probe in comparison with other instruments (FSSP's, Cloud Imaging Probes, holography instruments, etc.).

Page 8, Line 29 -31: You mention a new data acquisition software, but do you use analysis software to identify different kind of particles (plates, columns, etc.)? Do you analyze and sort the particles by hand or do you use some sort of algorithm?

**Minor comments**

Page 1, Line 12: change "form" to "from"

Page 2, Line 22: Here, you use a headline followed by another headline. It looks strange when there is a headline with no following content.

Page 2, Line 23: The paragraph about the "Trigger detector" is included in the subsection "Light scattering detection system". Do you think it is the right place? You should consider putting it before this subsection, because the Trigger also starts the image acquisition (see Page 7, Line 28-29). It means that the Trigger detector initiates the light scattering system and the imaging system.

Page 3, Line 8: Is "sensing area" similar to "sample volume"? If it is, change it.

Page 3, Line 9: "roughly" Can you deliver an uncertainty of these droplet diameters?

Page 3, Line 28: You might answer it in Part 1, but are the mirrors heated to avoid condensation?

Page 3, Line 31: You should mention in this sentence for what reason it is not feasible.

Page 4, Line 17 – 18: You show in Figure 2 the redesigned fiber-to-MAPMT coupler and the simulated irradiation. If you additionally show the simulations here before the redesign it would help to explain why you needed a redesign.

Page 8, Line 25: "several kHz" Be more specific.

Page 8, Line 31: "QuickUSB library and the library that comes with the camera" Give a reference.

Page 9, Line 42: "at least for cirrus cases" What is the typical particle distance between cirrus particles. Does it mean that you can not exclude shattering for liquid clouds?

Page 9, Line 27: You should cite the BAMS paper here concerning ML-CIRRUS
http://journals.ametsoc.org/doi/abs/10.1175/BAMS-D-15-00213.1

Page 11, Line 5: Change "The two imaged droplets with…" to "The two imaged droplets in Figure 12 with…"

Page 12, Line 11: Change "In a first analysis, bullet-rosettes were…" to "In a first analysis, bullet-rosettes (see Figure 14) were…"

Page 14, Line 1 – 6: As mentioned before, I think it is time for an intercomparison with other instruments.

Figure 2b: Legend is too small
Caption Figure 2: "nephelometer" not "nephlometer", ….the the …..

Figure 3: The numbers on the axes are too small

Figure 3 and 4: Stay consistent with the Figures. For Figure 2 you use "a" for the left and "b" for the right figure. Here you talk about "left", "right", "upper" and "lower" panel.

Figure 4: Left and right figure are inconsistent. Labels are different. Brackets around the units are different. Label size is different. Ticks for y-axis are different. The illustrations of the electrical crosstalk levels are different too.

Figure 16: Mark the images with numbers or letters. Then you can add these number to the lines of the angular scattering functions. Like in Figure 13.

All Figures: Stay consistent. Keep [unit] or (unit).

---

## Author Comment (AC1) · 29 Nov 2017

**We thank the anonymous reviewer for the helpful comments. These comments helped to substantially improve the manuscript. Below we give detailed answers to the individual reviewer comments in blue.**

This paper characterizes the response of the PHIPS-HALO probe that was introduced in the Part 1 paper published last year. In addition to the laboratory characterization of the light scattering detection system, imaging system and electronics, some first results from two field campaigns in the Arctic and in the vicinity of Colorado during the ARISTO 2017 field campaign are presented. I find that the probe does offer a unique way of looking at data from having stereoscopic images and from having scattering phase functions of individual particles coincident with the scattering properties, and will likely allow some fundamental questions describing the relation to cloud microphysics and radiation to be answered in the near future. Thus, I think its publication in AMT is appropriate. However, I am recommending a few minor revisions that I think will improve the quality and readability of the paper.

1) The paper discusses the PHIPS-HALO in isolation from any other imaging probes and scattering probes that are currently available. It would be nice to compare the advantages and disadvantages of PHIPS-HALO with some of these other probes. Perhaps a table could be constructed where some parameters of different probes that image particles and scattering phase functions could be compared (e.g., sample area, number of crystals imaged in given time period, range of particle sizes, sizes of particles detected, data volume, etc.). This would be helpful for future users of the probe.

We agree that future users of the probe might find it helpful to have a comparison of basic instrument parameters with other existing probes. However, a detailed comparison of PHIPS-HALO with the the Polar Nephelometer (PN) and the Cloud Particle Imager (CPI) is already given in Part 1 (Tables 1 and 2 therein). Further characterizations of the instrument have now quantified the size of the sensing area  $A_{sa}$ , which was not compared in Part 1. Therefore, we added the following paragraph to the Subsection "Trigger detector" of Section 2:

"The sensing area of PHIPS-HALO and, therefore, its volume sampling rate, is significantly smaller compared to other imaging probes (e. g. the Cloud Imaging Probe CIP (DMT, Boulder):  $A_{sa}=1.6 \text{ cm}^2$  or the Cloud Particle Imager CPI (SPEC Inc., Boulder):  $A_{sa}=0.04 \text{ cm}^2$ ), but is comparable to the sensing area of conventional single particle light scattering probes (e. g. the Cloud and Aerosol Spectrometer CAS (DMT, Boulder):  $A_{sa}=0.0025 \text{ cm}^2$  or the Fast Cloud Droplet Probe FCDP (SPEC Inc., Boulder):  $A_{sa}=0.0025 \text{ cm}^2$ ). The reason for this small

sensing area used in PHIPS-HALO is that angular light scattering functions are measured on a particle-by-particle basis for typical cloud situations up to 1000 particles per cm-3 (see the discussion of the coincidence characteristics below). The Polar Nephelometer (PN) instrument from Laboratoire de Météorologie Physique (LaMP), Université Blaise Pascal, Clermont-Ferrand, France (Gayet et al., 1997) uses a significantly larger sensing area of  $A_{sa}=0.5$ cm2. In contrast to PHIPS-HALO, the PN is constructed to measure the angular light scattering function of particle ensembles with the aim that scattering features related to single ice crystals and their specific orientations are averaged out (Gayet et al., 1997). A comparison of further parameters of PHIPS- HALO with the PN and the CPI are given in Tables 1 and 2 of Part 1, respectively."

2) I think that the authors should exhibit more caution in some conclusions that they make out of a very limited set of data. I think the current paper is very powerful at showing the types of questions the probes can answer, but less powerful at actually answering these questions given the very limited amount of data that are presented. Some of the conclusions in Sections 3.2 and 3.3.1 are especially problematic. For example, the authors claim that the images in Fig. 13 show that a highly structured crystal (b) gives flat and featureless phase functions, whereas less structured crystals (a) exhibit peaks at two specific angles. I found this less than convincing: when I compare the (a) and (b) images I only see that the image in (b) is darker, as later commented on by the author. If the b particle is indeed more structured, the authors need to show the specific places on the crystals (perhaps circled) where this structure is seen. I am also not convinced that the particles in (a) and (b) are similarly oriented. They look to me that they could be oriented with different angles. Can the authors do some simple scattering simulations to show how different orientations of the same particle affect the scattering phase function? If there is a difference in 10 degrees, for example, is this sufficient to show different scattering functions? Over what angles would the scattering patterns be similar, and how close do the scattering functions need to be in order to be classified as similar?

We agree with the reviewer that our conclusions are not always fully justified by these limited examples given in Section 3 "First results". This might be related to the excitement the authors experience when presenting these unique data from PHIPS-HALO.

In the specific case of the example given in Figure 13, we do not fully agree with the reviewer that a darker appearance of a non-absorbing object in a bright field micrograph doesn't tell anything about its structural complexity and, consequently, its spatial light scattering behavior, but acknowledge that further

detailed analyses including also light scattering simulations are necessary. We, therefore, addressed the reviewer concerns by rephrasing the paragraph in Sec. 3.2 describing the single particle light scattering results shown in Figure 13:

"Figure 13 gives single particle angular scattering functions measured for two plate-like ice particles during the ARISTO2017 project. These two plates were selected because (i) they have a similar size and (ii) they are similarly oriented, though the orientation of their c-axes differ by at least 10° in the horizontal plane. An inspection of the stereo images reveals that crystal (b) appears darker than crystal (a). A darker appearance of a non-absorbing object in bright field microscopy is the consequence of more object-air interface interactions of the light rays that incide and penetrate the object. This means that, in case of the ice crystals shown in Fig. 13, crystal (b) has likely more surface distortions in terms of steps, roughness, indentations, and air inclusions compared to crystal (a), which appears more transparent. As a consequence of this structural difference, the angular light scattering properties of the two crystals differ in terms of a higher fraction of diffuse light scattering (reflection) in case of the more structured crystal (b) compared to the less structured crystal (a). The corresponding measured angular light scattering functions of the two crystals, shown on the right side of Fig. 13, support this conclusion. Crystal (b) induces scattering intensities measured at the side- and backward directions that exceed those of the less structured crystal (a) by up to one order of magnitude. It is acknowledged, however, that detailed light scattering simulations, like in the work of Shcherbakov et al. (2006), are necessary to unambiguously prove that the observed differences can be attributed to differences in the ice crystal complexity."

Similarly, I am concerned with the analysis in Section 3.3.1 where the authors make the overarching claim that "particle ensembles composed of ice crystals that show a significant complexity on a single particle basis possess similar flat and featureless average angular scattering function even if their basic crystal habit differ (columnar vs. plate-like in this case)." I think a much more thorough analysis needs to be done, including using scattering models to see how different orientations of crystals and different constructions of bullet rosettes with varying numbers of rosettes and orientations, affect the scattering properties before making such a conclusion. The authors themselves seem to explicitly acknowledge this when they stated that the "above examples have demonstrated that this question can be addressed by measurements with PHIPS-HALO, [but] further detailed analyses with larger data sets are necessary to come to statistically significant conclusions." I would recommend toning down the earlier

statements, and supplying some scattering simulations, to better justify the discussions in the earlier part of the paper.

We agree with the reviewer that the conclusion made from the habit-specific averaged scattering functions presented in Sec. 3.3.1 are too far-reaching. We therefore rephrased the appropriate paragraph:

"Interestingly, by comparing the habit-specific averaged scattering functions of Figs. 14 and 15 it becomes obvious that these functions do not significantly differ but show a featureless and rather flat angular dependence. Whether this is a general feature of complex atmospheric ice particle ensembles or a coincidence of the two selected cases, requires a more thorough analysis with larger single particle data sets and including state of the art ice particle optical models that is certainly beyond the scope of this technical paper. However, the presented habit-specific analysis of single particle light scattering data demonstrates the potential of PHIPS-HALO to answer the question which microphysical property of ice clouds dominate their angular light scattering behavior – the crystal habit or the crystal complexity in terms of distortions, inclusions, and surface roughness. This will be the subject of future studies after PHIPS-HALO has been participated in further cloud related aircraft projects."

Minor Comments:

Page 3, line 1: How does the sample area (and other parameters) compare with other probes? See major comment 1.

Changed accordingly. See answer to major comment 1.

Page 8, line 21 bits not bit

Corrected.

Page 8, line 26: lose not loosing

Corrected.

Page 9, line 19, was not were

Corrected.

Page 9 line26: Remove was

Corrected.

Page 10, line 3: I don't think ARISTO was designed to test instrumentation from SOCRATES, even though many of the experiments used in ARISTO will ultimately be used in SOCRATES

**Agreed and rephrased.**

Page 10, lines 15-16: I don't think it is true that there are always portions of the particle in other imaging probes that are out of focus. While I think it is true that some portions of the particles imaged that are out of focus, but there are individual particles that are entirely in focus. I'm not sure if this is a misinterpretation of the English that is written, but it should be noted that entire particles are in focus in other probes (though some particles are entirely out of focus).

**Agreed and rephrased.**

Page 13, line 32: missions not mission

Corrected.

---

## Author Comment (AC2) · 29 Nov 2017

**We thank Marcus Klingebiel for his helpful comments. These comments helped to substantially improve the manuscript. Below we repeat his comments and give detailed answers in blue.**

The manuscript "PHIPS-HALO: the airborne Particle Habit Imaging and Polar Scattering probe – Part 2: Characterization and first results" is the second part of a study presenting a novel aircraft optical cloud probe. This part is focusing on the characterization and the first measurements from the PHIPS-HALO instrument.

The unique part of the PHIPS-HALO is the combination of a polar nephelometer and a stereo imager. Both components together allow for measurements of the microphysical properties and the appropriate angular light scattering function of single particles.

In this manuscript, the authors characterize the main components of the instrument (light scattering detection system, imaging system, electronics) and present some first results from two research campaigns. For example, the authors explain very clearly why and how they redesigned the fiber-to-MAPMT coupler inside the polar nephelometer in order to avoid optical crosstalk between adjacent channels of the MAPMT, which is an important factor for obtaining reliable measurements. For the imaging system, they introduce a correction method, which is used to correct the oversizing of smaller cloud particles in order to get adequate results.

All in all, the manuscript is very well written and has a clear structure. I would suggest the manuscript to be published after minor revision. This should address the following points:

**Major comments**

Page 9, Line 26 – 33: You mention that the instrument was used during four aircraft missions. I am wondering how the measurements from the PHIPS-HALO agree with particle measurements from other imaging instruments or another polar nephelometer. I think this is the biggest weakness of the manuscript, because the authors show results from only a single instrument. It would be a beneficial to know if the measurements of the angular scattering function are similar to measurements from another polar nephelometer. The comparison could be done by using homogenous cloud sections. If other instruments were not available on the aircraft, it might be possible to use cloud chamber studies for an instrument intercomparison.

We agree with the reviewer that showing intercomparisons with other cloud probes (especially for the polar nephelometer part of PHIPS-HALO) would be beneficial. Now, there is only one other airborne polar nephelometer existing; the PN from LaMP, Clermont-Ferrand, so occasions to do such an intercomparison are sparse. Therefore, we haven't had yet the possibility to compare the fully functioning PHIPS-HALO with the PN in the field, but did comparisons in the AIDA cloud chamber, though the improved fiber coupler haven't been implemented at that time. The result of this intercomparison is already published in a study by Schnaiter et al. (2016) on the origin of ice crystal complexity in cirrus clouds.

We added the following paragraph to Section 2.2 "Polar nephelometer" to summarise the outcome of this comparison:

"It has not yet been possible to compare the improved polar nephelometer of PHIPS-HALO with the aircraft approved Polar Nephelometer (PN). However, the predecessor PHIPS-HALO nephelometer with the old fiber coupler was compared with the PN for ice particle ensembles generated in cirrus simulation experiments in the AIDA (Aerosol Interactions and Dynamics in the Atmosphere ) cloud chamber (Fig. 7 of Schnaiter et al. (2016)). For this comparison the averaged angular scattering functions from PHIPS-HALO were corrected for channel crosstalk and channel sensitivity characteristics as described in Part 1. A reasonable agreement of both instruments were found with maximum deviations in the normalized scattering functions of less than 50%."

Page 10, Line 12 – 28: You point out the advantages of the stereo imager very clearly, but the advantages of the whole PHIPS-HALO instrument in comparison to other instruments is neglected. It would be nice to have a table or a paragraph which summarizes the advantages of this novel cloud probe in comparison with other instruments (FSSP's, Cloud Imaging Probes, holography instruments, etc.).

We think that the paper clearly demonstrates the unique character of PHIPS-HALO. The instrument is primarily designed to provide experimental data on the most fundamental link between the microphysical properties of real atmospheric ice particles and their angular light scattering function on a single particle basis.

Therefore it is hard and simply not justified to talk about advantages or disadvantages of PHIPS-HALO with respect to other cloud probes. We acknowledge that having a section in the paper titled "Advantages of a stereo imager" is not justified without presenting the disadvantages of the PHIPS-HALO imager (with respect to other probes) at the same time. We therefore changed the section title to "Stereo-Microscopic Image Examples" and rephrased the section:

"Before results of the correlated microscopic and angular light scattering measurements are presented, examples of the stereo imaging method are shown to document the quality and information content that can be expected from the PHIPS-HALO imagery acquired under flight conditions. It is important to emphasize here that the stereo imaging method is essential for the overall concept of PHIPS-HALO as it is the basis for the interpretation of single particle angular scattering functions. The method provides not only a three dimensional impression of the imaged particle, but gives also its orientation with respect to the scattering plane. Both information parts are necessary to represent the particle in optical models for simulating its angular light scattering function.

A general problem in two dimensional optical imaging of ice crystals – even in the case of real in focus optical microscopy like used in PHIPS-HALO - is that there are always parts of the particle obscured in the image that makes a representation of its 10 three dimensional geometric structure impossible. In Fig. 11 two examples of skeleton plates are depicted that were sampled by PHIPS-HALO during ACLOUD in ice precipitation underneath a mid-level cloud at temperatures between -10°C and -14°C. These examples nicely demonstrate how the stereo imaging method enhances the microphysical information that can be drawn from the PHIPS-HALO stereo-micrographs of individual ice crystals. The stereo image examples shown in Fig. 11 reveal that these crystals are actually composed of multiple stacked skeleton plates. In the example (b) three hexagonal plates are concentrically stacked along the basal facet, which becomes obvious by inspecting the image of CTA1 (left). Having only the image of CTA2 (right) available, the crystal would have been classified most likely as a single skeleton plate. Although, a stacked plate arrangement is identifiable in CTA2 of example (a), the one side-plane that is radiating in a different direction becomes visible only by imaging the crystal under a different viewing angle as in the case of CTA1. Note that a stereo imaging approach is also used in the 2D-S probe (2 Dimensional Stereo probe, SPEC Inc., Boulder) in which two independent shadowgraph images of the same particle are recorded at a viewing distance of 90°. The examples given in Fig. 11 also show that the enhanced bright field image clarity due to the use of incoherent and monochromatic light as documented in the laboratory versions (Abdelmonem et al., 2011; Schön et al., 2011) is achieved also under flight conditions."

Page 8, Line 29 -31: You mention a new data acquisition software, but do you use analysis software to identify different kind of particles (plates, columns, etc.)? Do you analyze and sort the particles by hand or do you use some sort of algorithm?

The new data acquisition software is just for the data acquisition and storage, i.e. no further image processing is conducted at this stage. The raw images are processed after flight by our in-house developed analysis software as described in Schön et al. (2011) for area, equivalent diameter, maximum and minimum dimension, aspect ratio, and roundness. A reference to the Schön et al. (2011) paper is given in the first sentence of section 2.3.1. A habit-specific classification of the imaged crystals are conducted by visual inspection of each stereo image.

We acknowledge this by adding "manually selected" to the second and third paragraph of section 3.3.1 "Habit-specific angular scattering functions from ice clouds".

**Minor comments**

Page 1, Line 12: change "form" to "from"

**Corrected.**

Page 2, Line 22: Here, you use a headline followed by another headline. It looks strange when there is a headline with no following content.

Section 2 has been reorganized. See our answer to the following comment.

Page 2, Line 23: The paragraph about the "Trigger detector" is included in the subsection "Light scattering detection system". Do you think it is the right place? You should consider putting it before this subsection, because the Trigger also starts the image acquisition (see Page 7, Line 28-29). It means that the Trigger detector initiates the light scattering system and the imaging system.

We agree with the reviewer that the trigger detector represents an important and independent part of the system and should be presented on the same level as the polar nephelometer and the imager parts. We therefore reorganized section 2, which has now the subsections "2.1 Trigger detector", "2.2 Polar nephelometer", and "2.3 Imaging system".

Page 3, Line 8: Is "sensing area" similar to "sample volume"? If it is, change it.

No, sensing area is not the same as sample volume here. However, we acknowledge that the terms were not consistently used in the discussion manuscript and revised the paper for a consistent use of the terms "sensing area", "sensing volume", and "volume sampling rate".

Page 3, Line 9: "roughly" Can you deliver an uncertainty of these droplet diameters?

Changed "roughly" to "77  $\pm 0.1 \mu$ m" as this is the exact result from the image analysis.

Page 3, Line 28: You might answer it in Part 1, but are the mirrors heated to avoid condensation?

All the optical components including the off-axis parabola mirrors are heated to avoid condensation. This is already mentioned in section 2.1 of Part 1:

"All optical components are heated to temperatures above the dew point to prevent water condensation on optics or ice aggregation which may clog the air path."

Page 3, Line 31: You should mention in this sentence for what reason it is not feasible.

As mentioned here this was already reasoned in Part 1. However, we slightly modified this sentence to:

"As reasoned in Part 1, the original concept idea of an additional 1° to 10° measurement at 1° resolution is not feasible with the actual set up, and, therefore, these channels are no longer used."

Page 4, Line 17 – 18: You show in Figure 2 the redesigned fiber-to-MAPMT coupler and the simulated irradiation. If you additionally show the simulations here before the redesign it would help to explain why you needed a redesign.

We think that the need for a redesign of the fiber coupler was satisfactorily reasoned in section "2.1.2 Polar nephelometer":

"This [residual optical] crosstalk could be clearly attributed to the fact that the numerical aperture (NA) and the diameter of the PMMA fibers were too large in combination with the minimum distance to the anode array of the MAPMT constrained by the 1.5 mm thickness of the MAPMT protection window. To solve this crosstalk problem the following redesign of the fiber-to-MAPMT coupler was performed."

Maybe the reviewer is confused by the term "redesign". Our line of argument here is: 1. The reason for the optical crosstalk was identified (see above). 2. A

**redesign** of the coupler in terms of the points (a) and (b) given in section 2.1.2 are envisaged. 3. To deduce the best distances between the fiber ends and the gradient index lenses and between the lenses and the MAPMT protection window, optical simulations have to be conducted. 4. The result of this simulations (distances with the best result in terms of crosstalk free light coupling) are used in the **mechanical design** and finally **manufacturing** of the new coupler.

To make this argumentation line more clear we changed the initial sentences of the second paragraph and third paragraph of section 2.2 "Polar nephelometer" to:

"Before a new coupler was manufactured based on this redesign considerations, comprehensive optical engineering simulations had been performed to define the optimal distances between the fiber ends and the index lenses as well as between the index lenses and the MAPMT protection window."

and

"The coupler was then manufactured according to the results of the optical engineering simulations and was characterized in the laboratory."

Page 8, Line 25: "several kHz" Be more specific.

**Changed to "13 kHz".**

Page 8, Line 31: "QuickUSB library and the library that comes with the camera" Give a reference.

**References are given.**

Page 9, Line 42: "at least for cirrus cases" What is the typical particle distance between cirrus particles. Does it mean that you can not exclude shattering for liquid clouds?

In cirrus clouds, the inter-particle distances are typically >  $10^{-2}$  m and, therefore, the histogram of the inter-particle arrival times will be bimodal in case particle shattering is occurring (see Korolev and Field, Atmos. Meas. Tech., 8, 761–777, 2015). At higher concentrations representing mixed-phase and liquid clouds, the inter-particle distances can be in the order of the minimum measurable distance defined by the instrument acquisition dead time and the true air speed, and a clear separation of the shattering events in the inter-particle arrival times histogram is no longer possible. To our knowledge the electronic dead time of PHIPS-HALO

is comparable to those of other cloud probes, so excluding particle shattering in mixed-phase and liquid clouds is a general problem.

Page 9, Line 27: You should cite the BAMS paper here concerning ML-CIRRUS

http://journals.ametsoc.org/doi/abs/10.1175/BAMS-D-15-00213.1

**Reference added.**

Page 11, Line 5: Change "The two imaged droplets with..." to "The two imaged droplets in Figure 12 with..."

**Changed.**

Page 12, Line 11: Change "In a first analysis, bullet-rosettes were..." to "In a first analysis, bullet-rosettes (see Figure 14) were..."

**Changed.**

Page 14, Line 1 - 6: As mentioned before, I think it is time for an intercomparison with other instruments.

See our answers to the major comments 1 and 2 above.

Figure 2b: Legend is too small

**Changed.**

Caption Figure 2: "nephelometer" not "nephlometer", ....the the .....

Changed.

Figure 3: The numbers on the axes are too small

Changed.

Figure 3 and 4: Stay consistent with the Figures. For Figure 2 you use "a" for the left and "b" for the right figure. Here you talk about "left", "right", "upper" and "lower" panel.

We agree and consistently use now (a) and (b) in the figures and the text describing the figures.

Figure 4: Left and right figure are inconsistent. Labels are different. Brackets around the units are different. Label size is different. Ticks for y-axis are different. The illustrations of the electrical crosstalk levels are different too.

Changed to the same graph style left and right.

Figure 16: Mark the images with numbers or letters. Then you can add these number to the lines of the angular scattering functions. Like in Figure 13.

We addressed this reviewer comment, but realised that the graph of the scattering functions is getting messy when labelling all functions. As a compromise we labeled only two ice scattering functions in the graph and the corresponding images.

All Figures: Stay consistent. Keep [unit] or (unit).

Revised to be consistent in all figures.